# Chitosan-Coated Flexible Liposomes Magnify the Anticancer Activity and Bioavailability of Docetaxel: Impact on Composition

**DOI:** 10.3390/molecules24020250

**Published:** 2019-01-11

**Authors:** Mohammed O. Alshraim, Sibghatullah Sangi, Gamaleldin I. Harisa, Abdullah H. Alomrani, Osman Yusuf, Mohamed M. Badran

**Affiliations:** 1Pharmacy Department, King Abdulaziz Medical City, Ministry of National Guard-Health Affairs, Riyadh 11426, P.O. Box 22490, Saudi Arabia; oalshreem@gmail.com; 2Faculty of Pharmacy, Northern Border University, Arar 91911, P.O. Box 840, Saudi Arabia; Sibghatullah.Sangi@nbu.edu.sa; 3Department of Pharmaceutics, College of Pharmacy, King Saud University, Riyadh 11451, P.O. Box 2457, Saudi Arabia; Harisa@ksu.edu.sa (G.I.H.); aomrani@ksu.edu.sa (A.H.A.); osmanhlb@yahoo.com (O.Y.); 4Kayyali Chair for Pharmaceutical Industry, Department of Pharmaceutics, College of Pharmacy, King Saud University, Riyadh 11451, P.O. Box 2457, Saudi Arabia; 5Department of Biochemistry, College of Pharmacy, Al-Azhar University, Cairo P.O. Box 11751, Egypt; 6Nanomedicine unit (NMU-KSU), College of Pharmacy, King Saud University, Riyadh 11451, P.O. Box 2457, Saudi Arabia; 7Department of Pharmaceutics, College of Pharmacy, Al-Azhar University, Cairo P.O. Box 11751, Egypt

**Keywords:** liposomes, chitosomes, docetatxel, in vito cytotoxicity, bioavailability

## Abstract

Flexible liposomes (FLs) were developed as promising nano-carriers for anticancer drugs. Coating them with chitosan (CS) could improve their drug delivery properties. The aim of this study was to investigate the physicochemical characteristics, pharmacokinetics behavior, and cytotoxic efficacy of docetaxel (DTX)-loaded CS-coated FLs (C-FLs). DTX-loaded FLs and C-FLs were produced via thin-film evaporation and electrostatic deposition methods, respectively. To explore their physicochemical characterization, the particle size, zeta potential, encapsulation efficiency (EE%), morphology, and DTX release profiles were determined. In addition, pharmacokinetic studies were performed, and cytotoxic effect was assessed using colon cancer cells (HT29). Various FLs, dependent on the type of surfactant, were formed with particle sizes in the nano-range, 137.6 ± 6.3 to 238.2 ± 14.2 nm, and an EE% of 59–94%. Moreover, the zeta potential shifted from a negative to a positive value for C-FL with increased particle size and EE%, and the in vitro sustained-release profiles of C-FL compared to those of FL were evident. The optimized C-FL containing sodium deoxycholate (NDC) and dicetyl phosphate (DP) elicited enhanced pharmacokinetic parameters and cytotoxic efficiency compared to those of the uncoated ones and Onkotaxel^®^. In conclusion, this approach offers a promising solution for DTX delivery.

## 1. Introduction

Docetaxel (DTX) is one of the most effective anticancer drugs, and was used over the past several decades against breast, ovarian, lung, colorectal, and head and neck cancers [1,2]. Owing to the poor solubility of DTX, its main marketed product that is used clinically, Taxotere^®^, consists of a large amount of Tween-80 (more than 1 g/mL), which is diluted with a provided solvent composed of 13% *w*/*w* ethanol in water [3]. Unfortunately, the utilization of ethanol and Tween-80 as a solvent mixture for DTX can yield several side effects and restrictions such as hypersensitivity, hemolysis, and lower uptake by cancer cells, leading to increased contact with other body cells [4,5]. Considerable work was focused on developing alternative, less toxic, and more successful formulations of DTX.

Liposomes received great attention as highly applicable nano-systems with properties such as biocompatibility, biodegradability, and less toxicity [6]. Moreover, liposomes are the most promising nano-carriers of anticancer drugs [6,7]. Many studies reported that flexible liposomes (FLs) are better carriers of drugs than conventional liposomes (CLs) [7]. In particularly, FLs have flexible membranes due to the existence of edge activators such as Tween-80 and sodium deoxycholate [8,9]. These vesicles have the ability to cross pores that are smaller in size than their size, because lipid bilayers possess a higher curvature than CLs [10]. Moreover, positive surface-coated liposomes were designed to improve liposomal characterizations such as stability, sustained drug release, and effective targetability [11]. In addition, these surface-modified liposomes could enhance drug cellular uptake due to their increased interaction with the cell surface [12].

Chitosan (CS) is a cationic polymer that possesses unique biological properties, including a bioadhesive effect, biocompatibility, biodegradability, and low toxicity [13]. CS is applied to improve muco-adhesiveness, which subsequently enhances drug absorption to promote sustained drug release [14]. CS is polycationic, and is used to form a positive layer around negatively charged liposomes via surface adsorption via electrostatic interaction [15]. This behavior of CS could enable efficient coating of phospholipid vesicles (chitosomes) in order to entrap drugs, leading to improved bioavailability [16]. Moreover, CS exhibits anticancer activities through interfering with the metabolism of tumor cells, thereby inhibiting cell growth [17]. Liposomes and chitosomes are also potential carriers of chemotherapeutic agents [7]. The positively charged liposomes might interact with the cell membrane, causing the opening of tight junctions, subsequently enhancing drug permeation [18]. However, CS-coated FLs based on the type of surfactant required to deliver DTX are yet to be described in the literature.

Therefore, the aims of this study were to fabricate DTX-loaded FL-T (flexible liposomes containing Tween-80), FL-T-DP (flexible liposomes containing Tween-80 and dicetyl phosphate (DP)), FL-NDC (flexible liposomes containing sodium deoxycholate), and FL-NDC-DP (flexible liposomes containing NDC and DP). Moreover, all aforementioned FLs were CS-coated, obtaining C-FL-T, C-FL-T-DP, C-FL-NDC, and C-FL-NDC-DP. The CS-CLs and uncoated CLs were prepared for comparison purposes. The prepared formulations were assessed in terms of particle size, zeta potential, polydispersity index (PDI), drug encapsulation efficiency (EE%), and in vitro DTX release, as well as in vivo pharmacokinetics and in vitro cytotoxicity.

## 2. Results and Discussion

The coated vesicles have the ability to extend the residence time of the drug in circulation, leading to high targeting effect. Moreover, CS was exploited as a chaperone for liposomes to govern their surface characteristics with large loading capacity [15]. Nano-sized properties of CS-coated liposomes showed an excellent drug delivery with non-immunogenic characteristics due to their biological nature [13,15]. In this study, FLs were fabricated to increase drug loading and the flexibility of the membrane depending on their composition (Table 1 and Table 2). DP and NDC were employed to induce a negative surface charge on the liposomes [9]. Therefore, CS-FL carriers were successfully prepared based on CS coating through electrostatic interactions onto a negatively charged liposomal surface. Nano-sized C-FLs could enhance the loading and delivery of DTX with high cytotoxic effect.

### 2.1. Particle Size and Zeta Potential

The particle size distribution and zeta potential of different formulations were assessed as shown in Table 3 and Table 4; all the investigated formulations were in the nanometer range. The particle sizes of liposomes were 238.2, 148.2, 174.6, 137.6, and 165.5 nm for CL, FL-T, L-T-DP, FL-NDC, and FL-NDC-DP, respectively. The obtained data revealed that FLs had a smaller particle size than CLs, which aligns with other publications [9,15,19]. The particle size of FLs was reduced by the presence of a non-ionic surfactant, which could enhance their elasticity with the formation of liposomes with a small radius [9,15]. However, the negatively charged FLs resulted in a larger particle size. This effect is due to the incidence of DP in FLs, which might change the spacing between the adjacent bilayers of FLs [20]. The negatively charged FLs might be attracted to the cationic drug (DTX) electrostatically, which may push apart head groups of phospholipids, thereby increasing vesicle diameter [20].

Subsequently, the CS-coated liposomes exhibited increased particle sizes of 328.6, 218.9, 284.1, 190.6, and 251.5 nm for C-CL, C-FL-T, C-FL-T-DP, C-FL-NDC, and C-FL-NDC-DP, respectively. As shown in Table 3 and Table 4, the increase in particle size was significant (*p* < 0.05) after CS coating, particularly for C-FL-T-DP and C-FL-NDC-DP, compared to the other coated liposomes. The chitosomal particle sizes were, therefore, larger than those of the liposomes; this was ascribed to CS adhesion to the surface of FLs, resulting in the formation of a CS layer [3]. These results are consistent with a previous report on CS-coated liposomes containing both coenzyme Q10 and α-lipoic acid [21]. In the present work, the values of PDI were within the range of 0.219 to 0.581 for both systems, indicating an acceptable degree of polydispersity [22].

Zeta potential values are a particularly important parameter as they affect liposomal stability [23]. Figure 1 represents the values of zeta potential for the liposomes and chitosomes. FLs had negative charges of −5.6, −12.8, −38.1, −23.4, and −41.8 mV for CL, FL-T, FL-T-DP, FL-NDC, and FL-NDC-DP, respectively. However, CS-coated liposomes exhibited positive charges of 9.7, 14.7, 36.8, 21.6, and 44.9 mV for C-CL, C-FL-T, C-FL-T-DP, C-FL-NDC, and C-FL-NDC-DP, respectively. The statistical significance for all zeta potential values was detected after CS coating (*p* < 0.05).

The results confirm the interaction between CS and liposomes via electrostatic attraction. The high zeta potential of the coated liposomes (C-FL-T-DP and C-FL-NDC-DP) may have resulted from the adsorption of large amounts of CS onto the liposomal surface. This behavior was caused by the presence of a negatively charged agent, DP, resulting in highly positive C-FLs. Additionally, CL had a neutral zeta potential (−5.6 mV), while C-CL showed a positive zeta potential value (9.6 mV). It was postulated that neutral liposomes are subjected to hydrogen bonding between their phospholipids and CS, producing positively charged liposomes [24].

### 2.2. Encapsulation Efficiency (EE%) and Drug Loading (DL%)

EE% and DL% are important factors in selecting drug-loaded liposomes. Owing to the hydrophobicity of DTX, it could be inserted into the lipid bilayer of liposomes during preparation. Herein, the coated liposomes displayed low levels of encapsulated DTX compared to uncoated liposomes [15]. As a result, notable differences were detected between different FLs and C-FLs, as shown in Table 3 and Table 4.

The EE% of FL-NDC-DP was 94.4%, which was higher than that of FL-NDC with 91.7%, compared to the EE% values of 83.8%, 72.3%, and 58.7% for the remaining uncoated liposomes. The values of DL% were 4.62%, 6.83%, 8.83%, 10.34%, and 11.85% for CL, FL-T, FL-T-DP, FL-NDC, and FL-NDC-DP, respectively. After coating, improvements in EE% and DL% were observed, with EE% values of 76.5%, 86.9%, 95.3%, 96.4%, and 98.6%, and DL% values of 6.05%, 7.96%, 9.81%, 11.06%, and 13.24% for C-CL, C-FL-T, C-FL-T-DP, C-FL-NDC, and C-FL-NDC-DP, respectively. Statistical significance for EE% and DL% was detected after CS coating in the case of C-CL, C-FL-T-DP, and C-FL-NDC-DP (*p* < 0.05). These results revealed that the chitosomes had a significantly higher EE% and DL% than liposomes. This action might be explained by surface properties following the addition of CS during preparation [15,25].

### 2.3. Morphological Characterization

Figure 2 displays the TEM images of FL-NDC-DP and C-FL-NDC-DP, which reveal the spherical structure of these formulations. After coating, some aggregations were observed; however, the common particles in the investigated formulations displayed a spherical shape with a rough shell. The CS coating did not affect the structure of the liposomes, demonstrating a thin layer enclosed on the membrane surface as shown previously [26].

### 2.4. In Vitro DTX Release Studies

It was reported that successful drug therapy can be governed by the residence time of the carriers in vivo [27]. Therefore, an in vitro release study may imitate a partial description of the residence time of a drug [28]. The in vitro release profiles of DTX from liposomes, chitosomes, and Onkotaxel^®^ were determined in phosphate-buffered saline (PBS; pH 7.4) at 37 °C, as displayed in Figure 3A,B. In the case of the control (Onkotaxel), a relatively rapid release of DTX was observed, with 70% released within 12 h and a nearly complete release (approximately 100%) within 72 h. This effect could be clarified by the high solubility of DTX in the presence of Tween-80 and ethyl alcohol, which produced the highest release amount of DTX [26]. Furthermore, DTX-loaded FLs and C-FLs displayed two phases, an early rapid release followed by a sustained release, unlike the control. Typically, the early quick release of DTX can be explained by its removal from the outer carrier surface. The sustained release results from drug diffusion from the lipid bilayer and the adhesive CS layer for chitosomes [29].

The liposomes exhibited high levels of DTX release at 12 h with 28.2%, 40.6%, 44.5%, 50.3%, and 62.5% for CL, FL-T, FL-T-DP, FL-NDC, and FL-NDC-DP, respectively. Furthermore, approximately 64.1%, 66.2%, 82.2%, 87.6%, and 93.6% DTX was released after 72 h for CL, FL-T, FL-T-DP, FL-NDC, and FL-NDC-DP, respectively. The continuous release of DTX from liposomes was due to the enclosed lipid layers, which allowed slow DTX release from the lipid matrix [14]. The remarkable released amount of DTX from FL-NDC-DP was due to its high EE% and the flexibility of the membrane in the presence of NDC [7]. Flexible liposomes have the ability to modify their shape to navigate through smaller openings [30]. Moreover, a higher amount of DTX released was detected in the presence of DP.

The CS-coated liposomes elicited more retardation of DTX release. CS, however, further improved the stability and sustained-release characteristics of the liposomes [31]. The values of DTX release from chitosomes (C-CL, C-FL-T, C-FL-T-DP, C-FL-NDC, and C-FL-NDC-DP) were 11.1%, 19.2%, 10.4%, 15.6%, and 9.2%, respectively, after 12 h (Figure 3). Furthermore, approximately 61.4%, 69%, 58.2%, 64.3%, and 53.8% DTX was released after 72 h from C-CL, C-FL-T, C-FL-T-DP, C-FL-NDC, and C-FL-NDC-DP, respectively. The sustained-release pattern of DTX from chitosomes compared to that from liposomes was attributed to the existence of the CS layer, which delayed the diffusion of DTX into the release medium [7].

### 2.5. Hemocompatibility Study

The biocompatibility of the prepared carriers was evaluated using the erythrocyte hemolysis test [14]. The results in Figure 4A,B show that incubating the erythrocytes with isotonic solution (negative control (NC)) maintained their integrity, resulting in no hemolysis. On the contrary, complete hemolysis was observed when erythrocytes were incubated with 10% Triton X100 (positive control (PC)). For the liposomal formulations (CL, FL-T, FL-T-DP, FL-NDC, and FL-NDC-DP), slight hemolysis as observed, which indicated good hemocompatabilty. Furthermore, minor-to-moderate hemolysis was observed with the chitosomes. The positively charged surface of these formulations was believed to be the cause of such action. The results of the hemocompatability test revealed that the hemolysis of erythrocytes was influenced by the surface charge of the formulations. Neutral and negatively charged particles did not result in any hemolysis and were safe toward erythrocytes. The greater positive zeta potential of the coated liposomes was the main cause of increased erythrocyte hemolysis [14].

In the present study, the increased hemolysis of the CS-coated FLs was due to the threshold of positivity charged vesicles (i.e., increased CS coating as indicated by the high value of zeta potential). However, uncoated FLs had a low extent of hemolysis due to absence of positive surface charge. Concerning C-FL-NDC-DP, high hemolysis (~27%) was exhibited due to increased positivity (44.9 mV) compared to C-CL (~8%), as presented in Figure 4. The effect of CS-coated FLs on erythrocyte hemolysis decreased upon decreasing the positive charge from 44.9 to 9.6 mV. It was reported that, upon increasing the zeta potential values of nanocarriers beyond 30 mV, increased hemolysis of erythrocytes could be detected [32]. Further studies are required to minimize the occurrence of hemolysis in formulations intended for clinical application.

### 2.6. In Vitro Cytotoxicity Studies

The cytotoxicity of FL-NDC-DP and C-FL-NDC-DP with or without DTX compared to Onkotaxel^®^ was evaluated using the 3-(4,5-dimethylthiazol-2-yl)-2,5-diphenyltetrazolium bromide (MTT) method. An MTT assay was performed with 5 μg/mL DTX incubated with HT-29 cells. Figure 5 represents the percentage cell viability of the HT-29 cells treated with the abovementioned DTX carriers. The unloaded formulations did not show significant cytotoxicity, with more than 94% of HT-29 cells surviving using the same concentration of DTX formulation. The percentage cell viability decreased after incubation with the formulations. Using FL-NDC-DP, 52% cell growth was displayed, which aligns with the in vitro release results. For the DTX control (Onkotaxel^®^), however, DTX was more efficient in the prevention of cell viability (65%), which could be attributed to the high release of DTX in the presence of Tween-80. Onkotaxel was more effective in preventing cell growth than the liposomes. In the case of C-FL-NDC-DP, cell viability was greatly reduced (80%). This influence was caused by the CS coating of the liposomes, which improved the cytotoxic effect of DX against HT-29. The positively charged surface of chitosomes provided an enhancement of DTX absorption [33]. Thus, C-FL-NDC-DP had a high cell attraction, which potentially increased the cellular uptake of DTX, leading to its cytotoxicity [14,34].

### 2.7. Pharmacokinetic Studies

A pharmacokinetic profile and parameters of Onkotaxel, FL-NDC-DP, and C-FL-NDC-DP were obtained for rats, as presented in Figure 6 and Table 5. A biphasic decline with an initial phase characterized by a rapid decline in DXT solution (Onkotaxel) and a slow decline in C-FL-NDC-DP was evident, unlike that seen in FL-NDC-DP (Figure 6). The half-life (t_1/2_) of C-FL-NDC-DP was 3.0-fold and 2.1-fold higher compared to Onkotaxel and FL-NDC-DP, respectively. The peak serum concentration (C_max_) of Onkotaxel after the administration of a single (intraperitoneal (IP)) bolus dose of 13 mg/kg was 5.6 mg/mL 30 min post injection, and was reduced to 0.197 mg/mL at 24 h.

The C_max_ of FL-NDC-DP at 30 min post injection was 17.1 mg/mL, and this value declined to 0.17 mg/mL at 24 h. Moreover, the C_max_ of C-FL-NDC-DP was 27.9 mg/mL at 30 min post injection, before declining to 0.24 mg/mL at 24 h. The area under the curve of DXT in C-FL-NDC-DP was 21.9-fold and 2.2-fold higher than that of Onkotaxel and FL-NDC-DP, respectively. In addition, this system resulted in increased mean residence time (MRT) from 3.5 h for Onkotaxel to 10.4 h for C-FL-NDC-DP (Table 5).

This effect indicated that C-FL-NDC-DP offered extended residence time. The relatively slow decline in serum concentration of C-FL-NDC-DP suggested a sustained-release profile, which was consistent with the results of the in vitro release study. The increased area under the curve and C_max_ of C-FL-NDC-DP correlated with the higher cytotoxic activity of DTX. Therefore, the coated CS layer could increase the sustained-release behavior of FL-NDC-DP and support its residence time in plasma [14].

## 3. Materials and Methods

### 3.1. Materials

Docetaxel (DTX) was purchased from Hangzhou Hyper Chemical Co., Ltd. (Zhejiang, China). Cholesterol, dicetyl phosphate, and 3-(4,5-dimethyl-thiazol-2-yl)-2,5-diphenyltetrazolium bromide (MTT) were purchased from Sigma-Aldrich Chemical Co. Ltd. (St. Louis, MO, USA). Lipoid S100 (phosphatidylcholine (PC), soybean lecithin, >94% PC) was purchased from Lipoid GmbH (Ludwigshafen, Germany). Sodium deoxycholate, methanol, and acetonitrile (HPLC grade) were purchased from Fisher Scientific Co., (Hampton, NH, USA). Tween-80 was obtained from BDH, Organics, (England, UK). All other chemicals were of analytical grade.

### 3.2. Preparation of DTX-Loaded Liposomes and Chitosomes

DTX-loaded FLs were produced using the thin-film hydration procedure as previously described [35]. In brief, PC, cholesterol, Tween-80, sodium deoxycholate (NDC), and dicetyl phosphate (DP) were dissolved in a 2:1 (*v/v*) chloroform:methanol mixture in a round-bottom flask. The organic solvents were rotary-evaporated under vacuum (Buchi Rotavapor R200, Buchi Architect Co., Ltd., Flawil, Switzerland) to form a dry lipid film; this was followed by flushing with nitrogen gas to completely remove the organic solvent. The dry film was rehydrated with PBS (pH 7.4) and vortexed for 30 min. The dispersions were probe-sonicated (Badnelin, Germany) for 2 min at an amplitude of 60% in an ice bath to achieve smaller-sized liposomes [35].

To prepare C-FLs, 1% (*w/v*) of the CS solution was obtained by dissolution in 0.5% (*v/v*) acetic acid solution (adjusted pH 5.5–6.0). The CS solution (1 mL) was then added dropwise to the DTX-loaded liposomes (1 mL) under probe-sonication for 2 min [34]. The resulting formulations were maintained under continuous stirring for 2 h to obtain C-FLs with 0.5% CS. All formulations were stored at 8 °C until use (Table 1 and Table 2). CLs and C-CLs were prepared for comparative purposes.

Different FLs and C-FLs, dependent on Tween-80, NDC, and DP content, were prepared to obtain formulations with the desired quality. To achieve successful formulations, the size, PDI, ζ potential, and encapsulation efficiency (EE) were qualified (Table 2 and Table 3).

### 3.3. Particle Size Distribution and Zeta Potential Measurements

Measurements of particle size and PDI of the formulations were performed using photon correlation spectroscopy with a Zetasizer Nano ZS (Malvern Instruments, Malvern, UK). Samples were diluted using a large volume of 0.2 µm filtered deionized water to prevent the multi-scattering phenomenon, and were equilibrated at 25 °C. Zeta potential was measured (Malvern Instruments, Malvern, UK) based on electrophoretic mobility after the proper dilutions were performed. The results were then presented as an average of five measurements for each sample. All results are presented as the average of triplicate measurements.

### 3.4. Determination of Encapsulation Efficiency (EE%) and Drug Loading (DL%)

The EE% and DL% values of DTX-loaded liposomes and chitosomes were determined via an indirect method using ultracentrifugation [25,36]. The amount of DTX in the supernatant as a free drug was obtained following centrifugation at 30,000 rpm for 30 min using an Optima™ Max-E, Ultra Cooling Centrifuge (Beckman Coulter, Pasadena, CA, USA). The DTX in the supernatant was analyzed by HPLC, and the values of EE% and DL% were calculated according to the following equations:(1)$$\mathrm{EE}\%\;=\;\frac{{\mathrm{DTX}}_{\mathrm{total}}\;-{\mathrm{DTX}}_{\mathrm{free}}}{{\mathrm{DTX}}_{\mathrm{toal}\;}}\;\times 100$$
(2)$$\mathrm{DL}\%\;=\;\frac{{\mathrm{DTX}}_{\mathrm{total}\;}-{\;\mathrm{DTX}}_{\mathrm{free}}}{\mathrm{Total}\;\mathrm{weight}}\;\times 100$$
Where *DTX* initially added was considered as *DTX*_total_ and the amount of *DTX* in the supernatant was *DTX*_free_.

### 3.5. Morphological Characterization

The morphology of the chitosan-coated DTX liposomes was visualized using transmission electron microscopy (TEM). A drop of DTX-loaded liposome (FL-NDC-DP) and chitosome (C-FL-NDC-DP) was deposited onto a copper grid and negatively stained with uranyl acetate solution (2% *w/v*). The excess solution was removed using a filter paper. The samples were air-dried for 30 min and observed by TEM (JEM-1011, JEOL, Tokyo, Japan) at 60 kV.

### 3.6. In Vitro DTX Release Studies

The in vitro release profiles of DTX from liposomes and chitosomes were assessed using the dialysis bag technique [33,34,37]. The formulations and Onkotaxel^®^ (~5 mg of DTX) were sealed in a pre-swelled dialysis bag (molecular weight cut-off (MWCO) 8kDa) and immersed in PBS (pH 7.4) containing 0.5% Tween-80 in a water-bath shaker (JULABO GmbH, Seelbach, Germany) with a shaking speed of 100 rpm at 37 ± 0.5 °C. At designated time intervals, 3 mL samples were withdrawn from the release media, and replaced with an equal volume of fresh release medium. The amount of DTX in each sample after 0.22 µm filtration was analyzed via HPLC. The experiment was performed in triplicate.

### 3.7. In Vitro Hemocompatability Study

The hemolysis study was used for screening purposes to provide initial feedback on the hemocompatability of the investigated nanocarriers. A lack of hemolysis is desired for the hemocompatibility of prepared nanocarriers. In the present study, a hemolysis test was performed to verify the safety of FLs and C-FLs using the blood of a healthy rat. The compatibility of the investigated liposomes and chitosomes with normal cells was estimated using a hemocompatability test. For this test, we adapted the protocol by Huang et al. [38] with minor modifications. In brief, an erythrocyte suspension (15 μL) and 1500 µL of phosphate buffer solution were placed in a test tube, along with 200 μL of the investigated formulation. After gentle mixing and standing for 2 h at room temperature, the sample was centrifuged at 1500 rpm for 5 min. A phosphate-buffered saline (PBS) solution was used as a negative control (NC; 0% hemolysis of erythrocytes) and 10% Triton X100 was used as a positive control (PC; 100% hemolysis of erythrocytes). Hemolysis was calculated using the following equation:(3)$$\mathrm{HR}\%\;=\;\frac{\mathrm{Absorbance}\;\mathrm{of}\;\mathrm{sample}-\mathrm{Absorbance}\;\mathrm{of}\;\mathrm{NC}}{\mathrm{Absorbance}\;\mathrm{of}\;\mathrm{PC}-\mathrm{Absorbance}\;\mathrm{of}\;\mathrm{NC}}\;\times 100$$

### 3.8. In Vitro Cytotoxicity Studies

The cytotoxicity of the loaded and unloaded FL-NDC-DP and C-FL-NDC-DP was evaluated and compared to Onkotaxel^®^ using an MTT colorimetric assay with human colon cancer (HT-29) cells [25]. The cells were seeded into 96-well plates at a density of 7 × 10^3^ cells/well and incubated at 37 °C under a humidified atmosphere of 5% CO_2_. The formulations and Onkotaxel^®^ were dispersed in PBS to reach 500 µg/mL DTX, and the mixture was diluted with Dulbecco’s modified Eagle medium (DMEM) to obtain 5 μg/mL DTX solution. Cells were then incubated for 48 h, and the remaining DTX was removed by careful washing with fresh DMEM. An MTT solution of 10 μL of 5 mg/mL PBS was added to cells and incubated for 4 h at 37 °C. The supernatant was then removed and solubilized in 50 μL of dimethyl sulfoxide (DMSO). The viability of the untreated cells was normalized to 100%, and the absorbance was measured at 540 nm in a microplate reader (ELX 800; Bio-Tek Instruments, Winooski, VT, USA). Data were expressed as the percentage of viable cells (treated) compared to the untreated cells of the control group.
(4)$$\mathrm{Cell}\;\mathrm{viability}\%\;=\;\frac{\mathrm{Untreated}\;\mathrm{cells}-\mathrm{Treated}\;\mathrm{cells}}{\mathrm{Untreated}\;\mathrm{cells}}\;\times 100$$

### 3.9. In Vivo Pharmacokinetic Study

To investigate the possible use of CS as a coating for DTX-loaded liposomes, an in vivo pharmacokinetic study of selected formulations was conducted. The method protocol was done according to the Research Centre Ethics of King Saud University, College of Pharmacy, Riyadh, Saudi Arabia (Ref. No.: KSU-SE-18-27). Wistar rats (300–350 g) were obtained from the Animal Care and Use Centre, according to the Research Centre Ethics of King Saud University, College of Pharmacy, Riyadh, Saudi Arabia (No.: KSU-SE-18-27). Rats were housed in plastic cages (each containing six rats) and acclimatized under standard laboratory conditions. The experimental guidelines of the animal care and use committee of King Saud University were strictly followed in this experiment. Rats were divided into three groups (six rats); Onkotaxel^®^- (control), FL-NDC-DP-, and C-FL-NDC-DP-treated groups were administered a dose equivalent to 13 mg/kg DTX via intraperitoneal (IP) injection [25,39]. All animals were fasted for 12 h before the start of the experiments. At 0.25, 0.5, 2, 4, 6, 8, 12, and 24 h after injection, 300 µL blood samples were collected from the orbital plexus into heparinized tubes.

Plasma samples were separated by centrifuging the blood at 6000 rpm for 5 min. Briefly, 30 µL of the internal solution (1 µg/mL paclitaxel) was added to 300 µL of plasma, followed by vortexing for 1 min. Samples were prepared by a single-step protein precipitation method using acetonitrile [25]. Plasma samples (300 μL) were added to 1.5 mL centrifuge tubes and vortexed for 1 min after adding 700 μL of acetonitrile. Samples were centrifuged at 13,000 rpm at 4 °C for 15 min. The supernatant was then transferred to HPLC vials and subjected to analysis. The proposed HPLC method was validated in the laboratory (unpublished data).

The pharmacokinetic parameters (PK) such as peak serum concentration (C_max_), time to reach peak serum concentration (T_max_), area under the curve from time 0 to *t* (AUC_0__→24_), area under the curve from time 0 to ∞ (AUC_0__→∞_), mean residence time (MRT), and elimination half-life (t_1/2_) were calculated using the PKsover add-in program for Microsoft Excel [40].

### 3.10. HPLC Analysis

DTX concentrations were determined using a reverse-phase HPLC method as described previously [41]. An HPLC system (Waters^TM^ 600 controller, Milford, MA, USA) equipped with a wavelength detector (Waters^TM^ 2487 a Dual λ Absorbance detector, Milford, MA, USA), pump (Waters^TM^ 1252 a Binary pump, Milford, MA, USA), and an automated sampling system (Waters^TM^ 717 Plus Autosampler, Milford, MA, USA) was used. The HPLC system was monitored by “Empower (Water)” software (Version 3, Empower Software Solutions Inc, Orlando, FL, USA). DTX was analyzed using a mobile phase consisting of a mixture of acetonitrile and water (55:45, *v/v*), filtered through a 0.45 µm-pore membrane. The mobile phase flowed through a reverse-phase C18 column (µ-Bondapak^TM^, 4.6 × 150 mm, 10 µm particle sizes, Waters, Milford, MA, USA) at a rate of 1 mL/min. The injection volume of each DTX sample was 20 μL, and the detection was achieved using an ultraviolet (UV) detector (Waters 2487, Milford, MA, USA) at 227 nm. All operations were carried out at room temperature.

### 3.11. Statistical Data Analysis

The results were expressed as means ± standard deviation (SD). Data analysis was performed via one-way ANOVA, followed by a Tukey-Kramer test for multiple comparisons. Data analysis was performed using GraphPad InStat software, Version 4 (GraphPad, ISI Software Inc., La Jolla, CA, USA). A 0.05 level of probability was used as the criterion for significance.

## 4. Conclusions

In this study, novel chitosomes loaded with DTX with a nano-range of vesicle sizes were successfully prepared depending on the surfactant used. The surfactant type played an important role in the effect displayed by liposomes and chitosomes. We observed that the presence of DP offered highly negatively charged liposomes, thus yielding liposomes with an evident CS coating. Hence, the CS-coated FLs could possess more advantages in drug delivery and targeting, particularly when used to transport anticancer agents. By comparing the data for EE%, DL%, in vitro drug release, and bioavailability, it was found that C-FL-NDC-DP displayed desirable results with high DTX loading, a sustained-release pattern, and enhanced pharmacokinetic parameters. A magnified anticancer activity by C-FL-NDC-DP toward HT-29 cells was observed. Therefore, C-FL-NDC-DP can be employed as a promising DTX nano-carrier for tumor targeting.

## Figures and Tables

**Figure 1 molecules-24-00250-f001:**
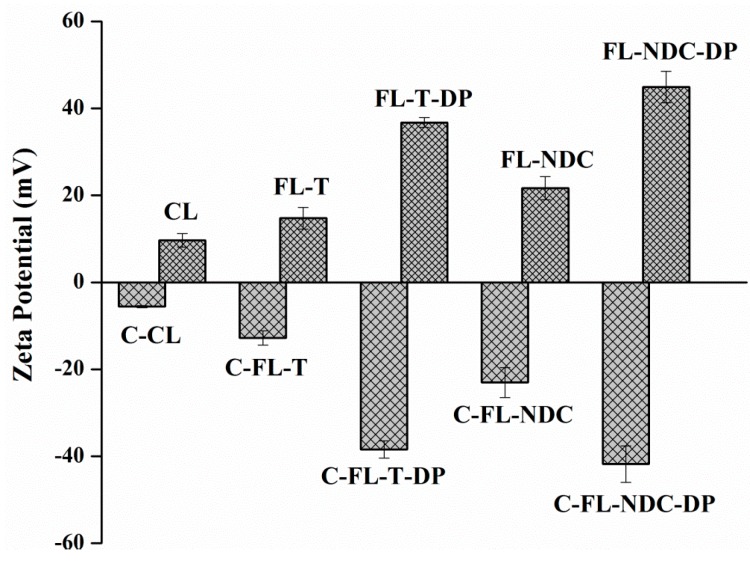
Zeta potentials of docetaxel (DXT)-loaded (5 mg/mL) liposomes and chitosomes. The data are expressed as means ± SD, *n* = 3. CL—conventional liposome; FL—flexible liposome; T—Tween-80; DP—dicetyl phosphate; NDC—sodium deoxycholate; C—chitosan-coated (0.5 mg/mL).

**Figure 2 molecules-24-00250-f002:**
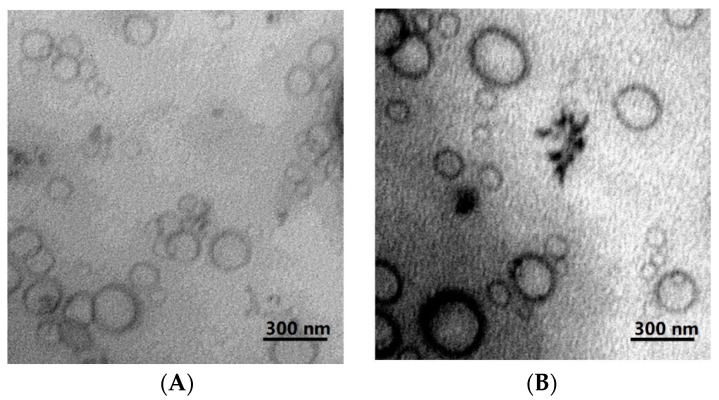
Transmission electron micrographs of FL-NDC-DP (**A**) and C-FL-NDC-DP (**B**).

**Figure 3 molecules-24-00250-f003:**
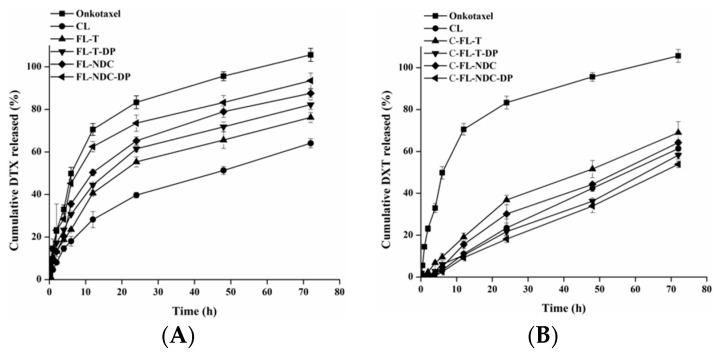
In vitro release profiles of DTX from Onkotaxel and liposomes (**A**) and chitosomes (**B**) in phosphate-buffered saline (PBS; pH 7.4) at 37 ± 0.5 °C. The data are expressed as means ± SD, *n* = 3.

**Figure 4 molecules-24-00250-f004:**
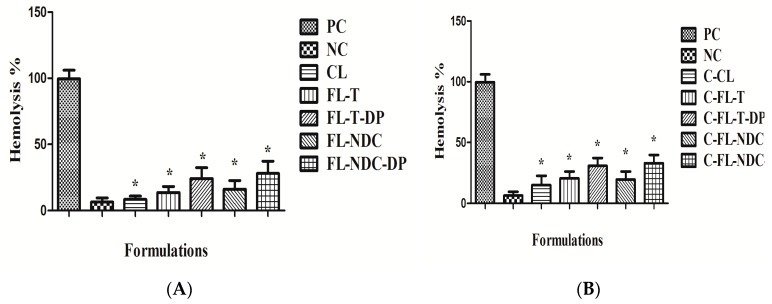
Hemolysis percentage of negative control (NC), positive control (PC), and liposomes (**A**) and chitosomes (**B**). The data are expressed as means ± SD, *n* = 3. * Significantly decreased from positive control at a *p* value ≤ 0.05.

**Figure 5 molecules-24-00250-f005:**
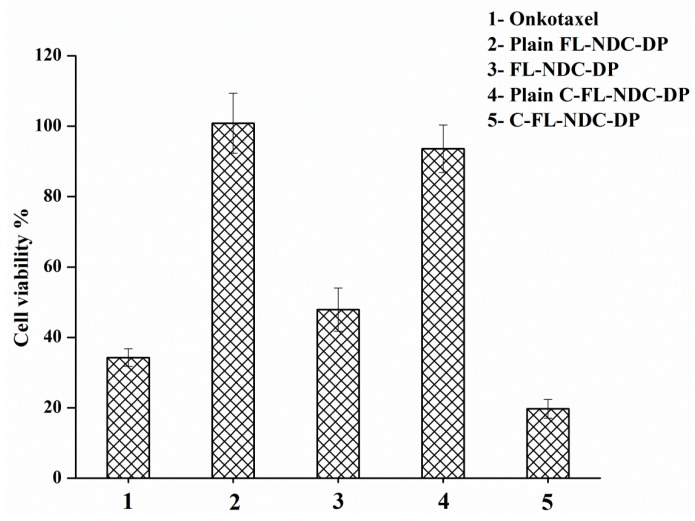
Cytotoxicity of DTX-loaded liposomes (FL-NDC-DP) and chitosomes (C-FL-NDC-DP) on the HT-29 cell line compared to Onkotaxel. The data are expressed as means ± SD, *n* = 3.

**Figure 6 molecules-24-00250-f006:**
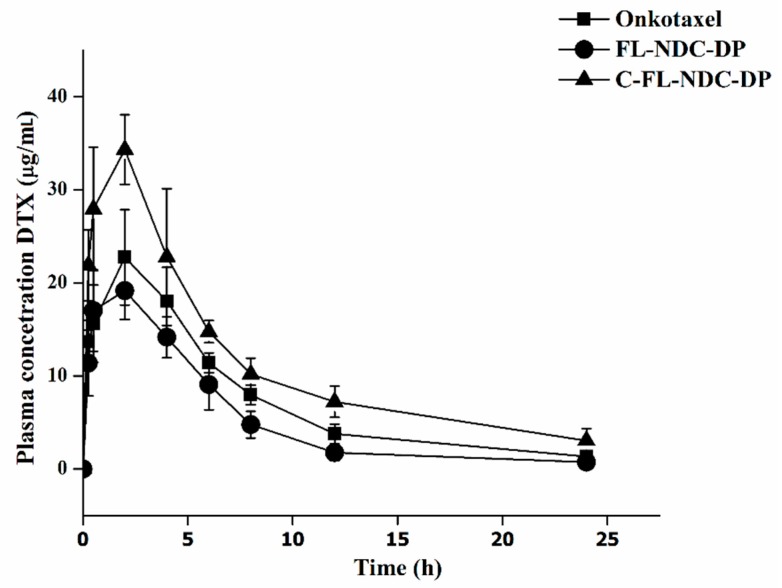
Plasma DTX concentration–time profile after intraperitoneal (IP) administration of FL-NDC-DP, C-FL-NDC-DP, and Onkotaxel to male Wistar rats at doses equivalent to 13 mg/kg (*n* = 6).

**Table 1 molecules-24-00250-t001:** The composition of docetaxel (DXT)-loaded liposomes. CL—conventional liposome; FL—flexible liposome; T—Tween-80; DP—dicetyl phosphate; NDC—sodium deoxycholate; CS—chitosan.

Codes Ingredients	CL	FL-T	FL-T-DP	FL-NDC	FL-NDC-DP
Lipid	0.9	0.9	0.9	0.9	0.9
Cholesterol	0.3	0.3	0.3	0.3	0.3
Tween-80	-	0.1	0.1		-
NDC	-	-	-	0.1	0.1
DP	-	-	0.1	-	0.1
CS (mg/mL)	-	-	-	-	-
DTX (mg/mL)	5	5	5	5	5

**Table 2 molecules-24-00250-t002:** The composition of docetaxel (DXT)-loaded chitosomes. C—chitosan-coated.

Codes Ingredients	C-CL	C-FL-T	C-FL-T-DP	C-FL-NDC	C-FL-NDC-DP
Lipid	0.9	0.9	0.9	0.9	0.9
Cholesterol	0.3	0.3	0.3	0.3	0.3
Tween-80	-	0.1	0.1	-	-
NDC	-	-	-	0.1	0.1
DP	-	-	0.1		0.1
CS (mg/mL)	0.5	0.5	0.5	0.5	0.5
DTX (mg/mL)	5	5	5	5	5

**Table 3 molecules-24-00250-t003:** Physicochemical characteristics of docetaxel (DXT)-loaded liposomes. PDI—polydispersity index; EE%—encapsulation efficiency; DL%—drug loading.

Codes	Particle Size (nm)	PDI	Zeta Potential (mV)	EE%	DL%
CL	238.2 ± 14.2	0.413 ± 0.030	−5.59 ± 0.25	58.7 ± 5.6	4.62 ± 0.84
FL-T	148.2 ± 10.7	0.282 ± 0.014	−12.77 ± 1.63	72.3 ± 4.5	6.83 ± 0.77
FL-T-DP	174.6 ± 8.1	0.224 ± 0.011	−38.43 ± 2.96	83.8 ± 2.7	8.83 ± 1.41
FL-SDC	137.6 ± 6.3	0.229 ± 0.028	−23.05 ± 3.46	91.7 ± 6.3	10.34 ± 1.27
FL-SDC-DP	165.5 ± 11.1	0.303 ± 0.072	−41.81 ± 4.20	94.4 ± 4.5	11.85 ± 1.35

These data are expressed as means ± SD, *n* = 3.

**Table 4 molecules-24-00250-t004:** Physicochemical characteristics of docetaxel (DXT)-loaded chitosomes.

Codes	Particle Size (nm)	PDI	Zeta Potential (mV)	EE%	DL%
C-CL	328.6 ± 9.1	0.581 ± 0.063	9.67 ± 1.56	76.5 ± 3.4	6.05 ± 0.78
C-FL-T	218.9 ± 4.3	0.251 ± 0.042	14.74 ± 2.49	86.9 ± 8.1	7.96 ± 0.68
C-FL-T-D	284.1 ± 11.5	0.283 ± 0.024	36.75 ± 1.46	95.3 ± 3.7	9.81 ± 1.05
C-FL-SDC	190.6 ± 9.4	0.219 ± 0.011	21.65 ± 2.68	96.4 ± 4.1	11.06 ± 1.21
C-FL-SDC-DC	251.5 ± 13.8	0.324 ± 0.077	44.92 ± 3.61	98.6 ± 7.2	13.24 ± 1.48

These data were expressed as means ± SD, *n* = 3.

**Table 5 molecules-24-00250-t005:** Pharmacokinetic parameters following intraperitoneal (IP) administration of the DTX formulations. C_max_—peak serum concentration; T_max_—time to reach peak serum concentration; t_1/2_—half-life; AUC—area under the curve; MRT—mean residence time.

Parameters	Onkotaxel^®^	FL-NDC-DP	C-FL-NDC-DP
Dose (mg/kg)	13	13	13
C_max_ (mg/L)	5.605 ± 1.322	19.415 ± 3.457	34.138 ± 4.752
T_max_ (h)	0.5	2	2
t_1/2_ (h)	3.067 ± 0.878	4.544 ± 1.217	9.288 ± 1.977
AUC_0__→t_ (mg/L.h)	13.429 ± 2.478	131.802 ± 7.331	294.287 ± 11.255
AUC_0__→∞_ (mg/L.h)	14.300 ± 2.115	136.563 ± 5.745	335.096 ± 14.427
MRT (h)	3.5	6.1	10.4

The data are expressed as means ± SD, *n* = 3.

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
