# Peer review of "Chitosan-Coated Flexible Liposomes Magnify the Anticancer Activity and Bioavailability of Docetaxel: Impact on Composition"

_molecules, 2019, doi:10.3390/molecules24020250_

Round 1

Reviewer 1 Report

This manuscript reported chitosan (CS) surface-modified FL (chitosomes) as an improved nano-carrier for DTX delivery. The authors evaluated the properties and application of C-FL compared with FL in terms of particle size, zeta potential, drug encapsulation EE%, and in vitro DTX release as well as in vivo pharmacokinetics and in vitro cytotoxicity. They demonstrated the coating with CS could improve their properties for drug delivery. The optimized C-F containing sodium deoxy cholate (NDC) and dicetyl phosphate (DP) elicited enhanced pharmacokinetic parameters and cytotoxic efficiency compared to uncoated ones and Onkotaxel. However, there are several issues for the system and for the manuscript that needs to be addressed.

1.      The order of tables is not well organized, and the Table 1, 2 should not be at the end of the manuscript.

2.      The methods section was put behind the results, and there is no description about CL FL-T, FL-T-DP, FL-SDC. The authors should interpret them when they first appear in the text.

3.      There are too many tables, so table 5 and 6 are suggested to merge into the Table 3 and 4.

4.      In Fig. 1, there seems some mistakes with CS-coated liposomes and FL. The CS-coated liposomes should exhibit positive charges according to the description in the text (line 109).

5.      The release study demonstrated the retarded drug release for the CS-coated liposomes, which is one of the advantages of the system. More studies are recommended to be done to demonstrate its other advantages, such as flow cytometry to evaluate the cellular uptake, and in vivo study to evaluate the anti-cancer effect.

6.      Although the CS-coated liposomes can retard drug release and may improve the cellular uptake compared to FL, more hemolysis was observed with CS-coated liposomes due to the positively charged surface, which is a disadvantage and major concern for further application.

Author Response

Dear Reviewer,

Thank you in advance for your valuable comments about our manuscript. We appreciate your comment, time and effort. Moreover, your comments are considered to improve our work.

1. The order of tables is not well organized, and the Table 1, 2 should not be at the end of the manuscript.

Response 1: Thank you for this comment about the organization of the tables in the manuscript.  All tables have been re-organized, specially tables 1 and 2.

2. The methods section was put behind the results, and there is no description about CL FL-T, FL-T-DP, FL-SDC. The authors should interpret them when they first appear in the text.

Response 2: We agree with the reviewer’s advice and have therefore revised the manuscript for the description of formulations codes. It has been clarified in the introduction section and throughout the article correctly.

3. There are too many tables, so table 5 and 6 are suggested to merge into the Table 3 and 4.

Response 3: The reviewer’s suggestion about tables 5 and 6 were obeyed and tables 3 and 4  were merged and cited in the text.

4. In Fig. 1, there seem some mistakes with CS-coated liposomes and FL. The CS-coated liposomes should exhibit positive charges according to the description in the text (line 109).

Response 4: The figure has been modified to fit the positive and negative charges of coated and uncoated liposomes. We apologize for these mistakes occur due deformation during data transfer and thank you for pointing out this error.

5. The release study demonstrated the retarded drug release for the CS-coated liposomes, which is one of the advantages of the system. More studies are recommended to be done to demonstrate its other advantages, such as flow cytometry to evaluate the cellular uptake, and in vivo study to evaluate the anti-cancer effect.

Response 5: The authors appreciate and thank the reviewer this recommendations to perform more studies to increase the manuscript quality. With respect to the reviewer’s request for a methodology section, MTT was firstly investigated for viability instead of actually measuring of proliferation. It measures the mitochondrial reductase activity, which represents the viability state of the cells. MTT is used as a sort of surrogate proliferation assay. As a continuation of this work (reviewer’s request), we are currently conducting a full mechanistic study, including the cellular uptake study by flow cytometry and in vivo antitumor activity for the same formulations. It will be published as a separate manuscript. The main concern in this manuscript is to check the feasibility of different types of chitosan-coated flexible liposomes containing DTX against HT29 cancer cell lines.

6. Although the CS-coated liposomes can retard drug release and may improve the cellular uptake compared to FL, more hemolysis was observed with CS-coated liposomes due to the positively charged surface, which is a disadvantage and major concern for further application.

Response 6: The disadvantage of this work is the hemolysis percent of chitosan coated liposomes is still high and the major concern for further application to decrease the hemolysis percent. The hemolysis study is used for screening purpose to provide initial feedback on the investigated nanocarriers. Then, the lack of hemolysis is directed for hemocompatibility of nanocarriers.

In the present study, the more hemolysis of the CS coated liposomes is due the threshold of positivity charged of the vesicles. Thus, the more CS coated as indicated by the high value of zeta potential.  However, uncoated liposomes have low extent of hemolysis due to absence of positive surface charge. C-FL-NDC-DP exhibited the highest hemolysis percent ( Ì´ 27%) due to more positivity (44.9 mV) as presented in Figure 4,  while C-CL showed low hemolysis ( Ì´ 8%). The effect of CS coated liposomes on erythrocyte hemolysis was decreased by decreasing of positive charges from 44.9 to 9.6 mV. It has been reported that the increasing of zeta potential than >30 mV, the interaction of nanocarriers with erythrocytes could be enhanced with high hemolysis (Yu et al., 2011). Therefore, the major concern for further application is avoiding-hemolysis is important for clinical application. The hemolysis was zeta potential dependent and results indicated that a low positive surface charge by the low CS concentration could lead to reduced hemolysis. For further application, more studies to evaluate the in vivo toxicity of CS coated liposomes are required to establish an in vitro and in vivo correlation for better prediction of toxicity in biological systems. Also, other various biocompatibility tests to improve the mechanistic understanding of the interaction between nanocarriers and erythrocytes are needed.

Yu, T.; Alexander, M.; Hamidreza, G. Impact of Silica Nanoparticle Design on Cellular Toxicity and Hemolytic Activity. ACSNANO, 2017, 5, 5717-28.

Reviewer 2 Report

This is an interesting study by Alsharim and colleagues on the effects of Chitosan coating of flexible liposomes loaded with cancer chemotherapeutic Docetaxel. There are several concerns with the manuscript which are being listed below as they appear in the manuscript.

1). The manuscript is poorly written and the English needs to be extensively revised. Please do not start sentences with "as well..." or "While...". The tense needs to be kept consistent across the manuscript. Poor grammar makes it hard to understand what the authors are trying to convey.

2). The use of abbreviations is not consistent and that is very confusing. For example CS-coated FL is abbreviated as C-FL but also C-F in the abstract. Are they one and the same??

3). There are too many abbreviations used and sometimes abbreviations of abbreviations as in above point #2 that makes it confusing to follow the manuscript.

4). The physicochemical properties of liposomes with and without Chitosan labeling in Table 3 and 4 need to be compared with statistical tests to show whether there the increase in size is significant or not. From looking at the numbers, i suspect that they will be but no statistical handling of the data is provided anywhere.

5). In the same tables the sample size or "N" needs to be provided.

6). Again, the changes in zeta potential in Figure 1 need to be compared statistically and p-values added if there is a significant difference.  

7). The encapsulation efficiencies and drug loading with and without Chitosan need to be statistically compared and p-values provided in Table 5 and 6.

8). Line 155 to 163 on page 8 needs to be typed double-spaced.

9). Why were in vitro release studies performed in PBS under otherwise physiological conditions? It would be closer to in vivo conditions if serum was utilized, or at  the very least cell culture medium instead of PBS. The experiments should be repeated in serum or at the very least a plausible explanation for use of PBS provided.

10). Convincing pharmacokinetic studies were performed in vivo in rats. However, the authors stop one step short of making this a really strong paper. No data/experiments were performed in vivo to show that while increasing the AUC with Chitosan labeling, there was also an increase in cancer response to this formulation in vivo, better tumor response, less metastasis, increased survival etc. in this study which would be the most useful outcome of this endeavor.

11). All Tables and Figures should be understandable in a stand alone manner. The abbreviations used in all these should have an explanatory legend at the bottom of each Table/Figures so the reader can make sense of them without having to read the whole article.

Author Response

Dear Reviewer, 

Thank you in   advance for your valuable comments about our manuscript. We appreciate your comment, time and effort. Moreover, your comments are considered to improve our work.

1). The manuscript is poorly written and the English needs to be extensively revised. Please do not start sentences with "as well..." or "While...". The tense needs to be kept consistent across the manuscript. Poor grammar makes it hard to understand what the authors are trying to convey.

Response 1: After revising our manuscript to address the reviewer comments, we have had rechecked. As a consequence, many minor grammatical and stylistic edits have been made throughout the text. We hope that this revised manuscript meets your expectations.

2). The use of abbreviations is not consistent and that is very confusing. For example CS-coated FL is abbreviated as C-FL but also C-F in the abstract. Are they one and the same?

Response 2: Thank you so much for catching this obvious and confusing error, which we have now corrected.

3). There are too many abbreviations used and sometimes abbreviations of abbreviations as in above point #2 that makes it confusing to follow the manuscript.

Response 3: The abbreviations were correctly checked and minimized.

4). The physicochemical properties of liposomes with and without Chitosan labeling in Table 3 and 4 need to be compared with statistical tests to show whether there the increase in size is significant or not. From looking at the numbers, I suspect that they will be but no statistical handling of the data is provided anywhere.

Response 4: The authors agree and appreciate this reviewer comment about the important of statistical tests in data presentation to be more accurate and precise. The reviewer advice was taken in consideration and done the revised manuscript.

5). In the same tables the sample size or "N" needs to be provided.

Response 5: We have provided N values in the manuscript.

6). Again, the changes in zeta potential in Figure 1 need to be compared statistically and p-values added if there is a significant difference. 

Response 6: Zeta potential statistical tests were done.

7). The encapsulation efficiencies and drug loading with and without Chitosan need to be statistically compared and p-values provided in Table 5 and 6.

Response 7: The encapsulation efficiencies and drug loading statistical tests were done.

8). Line 155 to 163 on page 8 needs to be typed double-spaced.

Response 8: The doubled-spaced lines in this page was achieved.

9). Why were in vitro release studies performed in PBS under otherwise physiological conditions? It would be closer to in vivo conditions if serum was utilized, or at  the very least cell culture medium instead of PBS. The experiments should be repeated in serum or at the very least a plausible explanation for use of PBS provided.

Response 9: The authors very much appreciate this helpful comment and agree that PBS is considered the most common medium for the in vitro release studies, but it needs a low ratio between the volume of the sample (nanocarriers) and medium, particularly for poorly water-soluble drugs like DTX to fit accuracy of the analysis. PBS is widely used for releasing a study without limitation of serum or PBS/serum mix due presence of proteins and other constituents that may interfere with drug analysis.  

To evade the limitation of analysis, drug remaining as nanocarriers in the system was measured at interval times to estimate the drug release indirectly.  The difference between the early and residual dose is reflected the released drug as long as the in vitro released drug remains stable in the medium. Furthermore, the serum-containing PBS may be a reasonable selection of in vitro release medium, which is closer to in vivo conditions that affects the release of the drug. Due to the solubilization of serum proteins, this medium are appropriate for a sink condition. But, DTX in serum-containing PBS requires a further extraction method to remove DTX from the proteins before analysis. The 0.5%tween/PBS is proper medium for DTX than serum/PBS, which does not need extra treatment for HPLC analysis, and provides a release profile with the fulfilled sink condition.  On the other hand, tween/PBS may not be well-suited to the dialysis bag method by using Tween/PBS as dispersion of nanocrriers in a dialysis bag. This is due to the released drug will be captured in the micelles of surfactant and not easily permit the membrane.

10). Convincing pharmacokinetic studies were performed in vivo in rats. However, the authors stop one step short of making this a really strong paper. No data/experiments were performed in vivo to show that while increasing the AUC with Chitosan labeling, there was also an increase in cancer response to this formulation in vivo, better tumor response, less metastasis, increased survival etc. in this study which would be the most useful outcome of this endeavor.

Response 10: We appreciate the reviewer’s comment and agree that in vivo anticancer activity is potentially important. In this study, we are now continue the performing in vivo mechanistic study for antitumor activity, including the cancer respond after induction and metastasis effect, when it is completed, it will be published as a separate manuscript. The main concern in this manuscript is to check the different flexible liposomes compared with their CS coating in correlation with overall anticancer activity.

11). All Tables and Figures should be understandable in a stand alone manner. The abbreviations used in all these should have an explanatory legend at the bottom of each Table/Figures so the reader can make sense of them without having to read the whole article.

Response 11: Thank you, we found your comments extremely helpful and have revised accordingly.

Round 2

Reviewer 1 Report

The manuscript becomes better after the authors' revision.

Author Response

Dear Reviewer,

Thank you in   advance for your thoughtful comments and efforts towards improving our manuscript. We appreciate your remarks, time and effort.

1. The manuscript becomes better after the authors' revision.

Response 1: We would like to thank the reviewer for the insightful review of  our manuscript.

Reviewer 2 Report

The authors have dealt with the comments that were easy to deal with (English language editing, better Table labels, correction of abbreviations, statistical tests etc.) and not done any new experiments like stability studies in serum that will be necessary as well as showing better tumor activity of the Chitosan labeled FL. I will accept their explanation that such studies are "Phase II" of this project and look forward to seeing those studies in print in the near future.

Only minor comments now are:

1). A line should be provided regarding what statistical tests were used, comparison against which control group and what post-hoc tests, if any, were performed and the statistical package used. I did not see that in the methods section.

2). Minor grammatical and punctuation mistakes still exist. Please correct.

3). In line 236, please give % symbol with each number. Also, what is 5,.3? That is obviously a typographical mistake. Please correct.  

Author Response

Dear Reviewer, 

Thank you in   advance for your valuable comments about our manuscript. We appreciate your comment, time and effort. Moreover, your comments are considered to improve our work.

1- A line should be provided regarding what statistical tests were used, comparison against which control group and what post-hoc tests, if any, were performed and the statistical package used. I did not see that in the methods section.

Response 1: Thank you for your valuable comments. The line about statistical tests was provide in the methods section.

2- Minor grammatical and punctuation mistakes still exist. Please correct.

Response 1: The whole manuscript has been revised and edited by an English-Expert, so we hope it now matches the your expectations.

3- In line 236, please give % symbol with each number. Also, what is 5,.3? That is obviously a typographical mistake. Please correct.  

Response 2: Thank for your comments. Accordingly, we have checked and corrected the errors.